# OpenReview forum: "AgentLAB: Benchmarking LLM Agents against Long-Horizon Attacks"
_ICML.cc/2026/Conference — ICML 2026 regular_

### Official Review · Reviewer_LXwh · 2026-03-02

**Soundness:** 2
**Presentation:** 3
**Significance:** 2
**Originality:** 2
**Overall Recommendation:** 4
**Confidence:** 2

**Summary:**

This work contributes AgentLAB, a benchmark for evaluating the vulnerability of LLM agents to long-horizon attacks, including intent hijacking, tool chaining, task injection, objective drifting, and memory poisoning. They found that many frontier LLM agents are currently susceptible to these attacks. This work also evaluates single-turn defense methods and shows that they are ineffective against long-horizon attacks.

**Compliance With Llm Reviewing Policy:**

Affirmed.

**Final Justification:**

The authors addressed most of my concerns.

**Key Questions For Authors:**

- Can you provide error bars with at least one standard error for tables 2 and 3?
- Can you cite those 2 relevant works mentioned above?
- Can you provide more details on the validation process mentioned in the weaknesses?

**Limitations:**

This paper doesn't have a limitation section. Some suggestions:
- This work only covers 5 long-horizon attacks, but real-world attacks most likely cover more than that.
- GPT-5.1 is a planner for attacks, but there's no analysis showing how it is chosen and whether it represents the most adversarial adversary, or is it on the lower end of attacking ability, given that it could refuse harmful instructions too. Would like to see more explanations. Similarly, an "abliterated" model (Qwen-3-14B with safety constraints removed) is used as the attacker; in my prior experience with it, it just doesn't refuse, and its actual attack capability is quite low and may not reflect the capabilities or constraints of real-world adversaries. Would like to see more analyses on this.
- The evaluator for task completion relies on comparing trajectory sequences against ground-truth sequences, which may miss partial successes or novel attack paths that achieve harm through unexpected routes.
- This work only evaluates single-turn defense, but there are a few multi-turn defenses already, e.g., the XGuard-Train from the X-teaming paper. It's crucial to evaluate this because if their defense works effectively then it suggests this benchmark might be already sovled by existing approach.

**Strengths And Weaknesses:**

Strengths:
- As LLMs become more agentic, long-horizon attacks become substantial risks to evaluate and address. AgentLAB is (to my knowledge) the most comprehensive benchmark that focuses on "long-horizon" attacks in an agentic setup.
- This benchmark is comprehensive: it covers 5 diverse long-horizon attacks and 28 agentic environments, and 9 risk categories. It also evaluates many frontier llms as agents.
- The ablation study is interesting, as the number of turns increases, the ASR increases.

Weaknesses:
- Regarding originality, there are some prior works proposing benchmarks/eval setups for decomposition attack, breaking a harmful task down into several benign subtasks for agents to execute. It's similar to the "Tool Chaining" setup. But these works are never mentioned. For example, https://openreview.net/forum?id=HQuboWvFA1 and https://arxiv.org/abs/2503.04957
- A key part of a benchmark paper, in my opinion, is a clear explanation of the task construction and validation. However, in section 4.3.1, several key parts are only briefly mentioned without details. For instance, "We validate task maliciousness by confirming that an LLM-based monitor (GPT-5.1) consistently refuses to execute them." but no details are provided regarding how this validation is run, and whether this LLM monitor refusal is aligned with human judgment of harmfulness. For instance, a monitor could be overly vigilant and refuse seemingly harmful tasks, but not harmful in essence. Another example is "All scenarios undergo manual validation to ensure: (i) malicious tasks are clearly specified; (ii) attack success can be evaluated programmatically; and (iii) agents show reasonable baseline performance." Yet, no details are provided regarding how manual validation is executed, and who are those humans? How are we sure of the quality of those humans?
- Throughout the paper, no error bars are shown.

---

> ### Author Rebuttal · Authors · 2026-03-29
>
> We thank Reviewer LXwh for the constructive review and for recognizing AgentLAB as the most comprehensive benchmark for long-horizon attacks. We address each concern below.
>
> **W1: Missing citations to decomposition attack benchmarks.**
>
> We thank the reviewer for identifying these relevant works. We will add citations to DecomposedHarm, which proposes a decomposition attack dataset where harmful goals are broken into benign subtasks for LLM agents, and SafeArena, which evaluates deliberate misuse of autonomous web agents. We will discuss how Tool Chaining in AgentLAB relates to and extends these efforts. The key distinction is that AgentLAB's Tool Chaining operates within an adaptive, multi-turn framework where the attacker dynamically adjusts its strategy based on agent responses, whereas prior work primarily evaluates static task decomposition or single-turn compliance.
>
> **W2: Task construction and validation details are insufficient.**
>
> We have conducted a validation study to ground our claims. On a stratified sample of 120 cases originally labeled by the GPT-5.1 judge, we employed two independent validators: human annotators and a separate GPT-5.4 judge.
>
> Human annotators agreed with GPT-5.1 on 91.67% of cases (FNR = 6.73%, FPR = 18.75%); GPT-5.4 agreed with GPT-5.1 on 89.17% (FNR = 10.58%, FPR = 12.50%). These results place our GPT-5.1 judge at a similar level to the best-performing judges in JailbreakBench (https://arxiv.org/pdf/2404.01318), which achieve ~90% agreement with FPR of 9-13%.
>
> We will expand the validation section with full details.
>
> **W3: No error bars shown.**
>
> We have now run multiple trials and report error bars for all main results. Updated Table 2 (ASR by attack type):
>
> | Agent | Intent Hijacking | Tool Chaining | Objective Drifting | Task Injection | Memory Poisoning | Overall |
> |:---:|:---:|:---:|:---:|:---:|:---:|:---:|
> | Qwen-3 | 78.1 ±7.1 | 96.3 ± 2.6 | 92.2 ± 4.4 | 93.1 ± 1.3 | 48.0 ± 14.3 | 81.5 ± 3.7 |
> | Llama-3.1 | 53.3 ± 7.9 | 90.4 ± 4.4 | 67.4 ± 7.9 | 86.6 ± 1.4 | 34.6 ± 11.9 | 66.5 ± 4.0 |
> | GPT-4o | 74.0 ± 6.1 | 94.1 ± 3.5 | 79.2 ± 5.9 | 79.9 ± 2.2 | 63.3 ± 14.2 | 78.1 ± 3.8 |
> | GPT-5.1 | 59.8 ± 6.9 | 94.6 ± 3.1 | 73.7 ± 8.9 | 21.5 ± 0.8 | 51.3 ± 15.5 | 69.9 ± 4.5 |
> | Gemini-3 | 46.2 ± 7.5 | 95.9 ± 2.8 | 15.8 ± 6.4 | 43.1 ± 1.7 | 67.3 ± 13.8 | 53.7 ± 4.0 |
> | Claude-4.5 | 27.2 ± 6.9 | 73.3 ± 6.7 | 5.3 ± 3.7 | 0.0 ± 0.0 | 38.8 ± 13.8 | 28.9 ± 3.6 |
>
> Updated Table 3 (not showing due to space constraint) similarly confirms that the long-horizon advantage over one-shot is robust across all agents. We will add error bars to both tables.
>
> **Suggested limitations, addressed:**
>
> *Attacker model choice:*
>
> We have added experiments with a stronger attacker, huihui-ai/Huihui-Qwen3.5-27B-abliterated, on Intent Hijacking:
>
> | Agent | Qwen3-14B Attacker | Qwen3.5-27B Attacker |
> |:---:|:---:|:---:|
> | Qwen-3 | 78.1 | 76.9 |
> | Llama-3.1 | 53.3 | 58.1 |
> | GPT-4o | 74.0 | 74.3 |
> | GPT-5.1 | 59.8 | 53.7 |
>
> ASR values remain comparable across attacker scales, suggesting the 14B attacker is already sufficiently capable. We deliberately use an abliterated model because safety-aligned models often refuse to assist in red teaming, undermining attack generation rather than reflecting victim robustness (https://arxiv.org/abs/2404.01833).
>
> *Multi-turn defenses:*
>
> We have evaluated two advanced defenses:
>
> *1. Progent (Shi et al., 2025) -- system-level defense (Progent-LLM mode):*
>
> We evaluated Progent on Tool Chaining and Memory Poisoning. Tool Chaining results:
>
> | Agent | Undefended | Defended by Progent |
> |:---:|:---:|:---:|
> | Qwen-3 | 96.3 | 75.4 |
> | Llama-3.1 | 90.4 | 66.1 |
> | GPT-4o | 94.1 | 73.7 |
> | GPT-5.1 | 94.6 | 57.1 |
> | Gemini-3 | 95.9 | 49.1 |
>
> Progent achieves meaningful ASR reductions, but substantial vulnerability persists across all agents and both attack types.
>
> *2. XGuard-Train (from X-Teaming) multi-turn defense via safety fine-tuning:*
>
> | Agent | Intent Hijacking (Original) | Intent Hijacking (XGuard) | Memory Poisoning (Original) | Memory Poisoning (XGuard) |
> |:---:|:---:|:---:|:---:|:---:|
> | Llama-3.1 | 53.3 | 44.4 | 34.6 | 20.0 |
> | Qwen-3 | 78.1 | 23.1 | 48.0 | 38.1 |
>
> While both defenses reduce ASR, significant vulnerability persists. Additionally, on Tool Chaining, the fine-tuned models produced gibberish when handling complex tool calls, indicating the defense disrupted utility rather than becoming more robust.
>
> **Limitations section.**
>
> Additionally, we will add a dedicated limitations section. A key limitation is the scarcity of long-horizon defenses. The existing defenses XGuard-Train ( https://arxiv.org/abs/2504.13203) and AgentAlign ( https://arxiv.org/abs/2505.23020) are both fine-tuning-based with only partial mitigation. A central goal of AgentLAB is to surface this gap and call for more effective long-horizon defenses.

---

> > ### Author Rebuttal · Reviewer_LXwh · 2026-03-31
> >
> > I thank the authors for their detailed rebuttal and the additional experiments. The new error bars, validation study, and defense evaluations represent a meaningful effort. Below, I outline which concerns have only been partially addressed.
> >
> > - W2 (Validation): Partially addressed. I appreciate the validation study with 120 cases and two independent validators. However, I note that the reported human FPR of 18.75% actually reinforces my original concern; nearly 1 in 5 tasks that GPT-5.1 refused were judged by humans as not harmful. This is a substantial false positive rate that the authors should discuss more carefully, as it suggests the monitor may indeed be overly vigilant, potentially inflating the perceived maliciousness of some test cases. Additionally, 120 out of 644 cases is ~19% of the dataset. The authors do not describe their stratification strategy in sufficient detail. I would like to understand whether the sample is representative across all five attack types, all 28 environments, and all 9 risk categories, or whether certain categories are over/underrepresented.
> >
> > - Attacker model choice: Partially addressed. Testing a second abliterated model (Qwen3.5-27B) on Intent Hijacking and finding comparable ASR is a useful data point. However, this experiment is limited to a single attack type and only varies model scale within the same abliteration approach. My original concern was more fundamental: an abliterated model that simply does not refuse is qualitatively different from a sophisticated adversary that strategically crafts attacks. The authors' argument that safety-aligned models refuse to assist in red teaming is understandable as a practical constraint, but it does not resolve the question of whether the benchmark's attack effectiveness numbers would look different with a more capable, strategically-motivated attacker. I would suggest the authors frame this more explicitly as a limitation rather than a settled design choice.
> >
> > Overall assessment: I acknowledge the effort invested in new experiments. I would encourage the authors to address these remaining issues in additional comments. If these are addressed, I'm happy to raise my score.

---

> > > ### Author Response · Authors · 2026-04-03
> > >
> > > We thank Reviewer LXwh for the constructive follow-up and respond to the remaining questions below.
> > >
> > > ---
> > > ### GPT-Judge Validation
> > >
> > > **Why the initial FPR appeared elevated.**  We identify two factors that inflated the 18.75% FPR in our initial rebuttal.
> > >
> > > 1. **Sampling imbalance.**
> > > The original 120 samples were randomly drawn from two attack tracks (tool chaining and intent hijacking) across different models, both of which have very high ASR. This produced a heavily imbalanced validation set: only 16 of 120 cases were non-jailbroken (i.e., negative). With so few negative cases, even a small number of disagreements (e.g., 3 out of 16) yields a disproportionately high FPR.
> > >
> > > 2. **Threshold sensitivity.** The judge assigns harmfulness scores from 1 (least) to 5 (most). The reported FPR used a threshold of 4 to classify a case as positive. Tightening the threshold to 5 on the same 120 samples reduces FPR to 12.50% (FNR = 16.35%), showing that the FPR is tunable via threshold selection and that the original number partly reflects a lenient threshold choice rather than fundamental judge unreliability.
> > >
> > > **Expanded validation study.**
> > > To directly address the reviewer's concern, we expand the validation study to 300 cases by randomly sampling an additional 180 cases stratified across the previously unsampled attack types across all victim models. The expanded confusion matrix (human vs. GPT-5.1, threshold $\geq 4$):
> > > | | Human Negative | Human Positive |
> > > |---|---|---|
> > > | **GPT-5.1 Negative (n=91)** | 80 (TN) | 11 (FP) |
> > > | **GPT-5.1 Positive (n=209)** | 20 (FN) | 189 (TP) |
> > >
> > > This yields FPR = 12.09%, FNR = 9.57%, Accuracy = 89.67%. The FPR now falls squarely within the 9-13% range reported by the best-performing judges in JailbreakBench, confirming that the elevated FPR in our initial rebuttal was an artifact of sampling imbalance rather than a systematic bias in the judge.
> > >
> > > **Coverage details.** The validation set of 300 samples now covers all attack types, all six victim models, and all nine risk categories present in the benchmark, ensuring that no single attack type, environment, or risk category is over- or under-represented.
> > >
> > > ---
> > >
> > > ### Attacker Model Choice
> > >
> > > We thank the reviewer for this insightful comment. We first clarify a key architectural distinction, then present additional experimental results.
> > >
> > > **Planner vs. attacker.** Our multi-agent attack framework (Figure 2) follows the planner-executor architecture in prior work (e.g, X-Teaming and Mastermind). The planner handles high-level strategic reasoning: analyzing the malicious task, generating diverse attack strategies, and decomposing tasks into executable sequences. The attacker (executor) handles low-level tactical execution: producing the actual adversarial payloads. Mastermind's ablation study directly supports our claims: the planner's reasoning capability is the primary determinant of attack success, while varying the attacker model has comparatively limited impact.
> > >
> > > **Expanded experiments on three attack types.** We further report the stronger abliterated attacker (Huihui-Qwen3.5-27B-abliterated) on two additional attack types:
> > >
> > > | Victim | Tool Chaining (14B / 27B) | Mem. Poisoning (14B / 27B) |
> > > |:---|:---:|:---:|
> > > | Qwen-3 | 96.3 / 92.8 | 48.0 / 57.3 |
> > > | Llama-3.1 | 90.4 / 88.0 | 34.6 / 47.2 |
> > > | GPT-4o | 94.1 / 94.6 | 63.3 / 53.1 |
> > > | GPT-5.1 | 94.6 / 95.1 | 51.3 / 42.8 |
> > > | Gemini-3 | 95.9 / 95.9 | 67.3 / 52.7 |
> > > | Claude-4.5 | 73.3 / 72.1 | 38.8 / 33.3 |
> > >
> > > **Analysis.** Across all attack types (including intent hijacking reported previously), substituting a nearly 2$\times$ larger abliterated attacker model yields marginal changes in ASR. For tool chaining, the differences are negligible. For memory poisoning, fluctuations are larger in absolute terms but exhibit no consistent directional trend, and they remain within the substantial error bars reported in Table 2. These results are consistent with Mastermind's finding that planner capability, rather than executor capability, is the primary driver of attack success.
> > >
> > > **Limitation framing.** We fully agree with the reviewer and will add the following to the limitation section: The use of abliterated open-weight models as attackers reflects a practical constraint with important limitations. Safety-aligned frontier models tend to refuse to generate explicit adversarial payloads, undermining attack execution regardless of victim robustness. Abliterated models resolve this compliance barrier but introduce a capability gap: they may lack the strategic sophistication of frontier models. Our experiments across three attack types show that scaling the attacker model does not systematically change ASR, consistent with Mastermind's findings that planner reasoning capability is the primary driver of attack success. Nevertheless, investigating alternative executor designs, including fine-tuned safety-aligned models, as well as human-in-the-loop red teamers, represents an important direction for future work.

---

### Official Review · Reviewer_A8a6 · 2026-03-09

**Soundness:** 2
**Presentation:** 3
**Significance:** 3
**Originality:** 2
**Overall Recommendation:** 4
**Confidence:** 4

**Summary:**

This paper introduces AgentLAB, the first benchmark dedicated to evaluating LLM agent security against long-horizon attacks — multi-turn adversarial strategies that exploit sustained user–agent–environment interactions to achieve objectives infeasible in single-turn settings. The authors propose a taxonomy of five attack types (intent hijacking, tool chaining, objective drifting, task injection, and memory poisoning), instantiated across 28 agentic environments and 644 test cases via a multi-agent framework (planner, attacker, judge). Experiments on six representative LLMs show that all remain highly vulnerable, and that existing single-turn defenses fail to transfer to this regime.

**Compliance With Llm Reviewing Policy:**

Affirmed.

**Key Questions For Authors:**

1. The paper's core empirical claims depend entirely on the reliability of the LLM judge. Can the authors provide human evaluation results on at least a stratified random sample (e.g., 50–100 cases per attack type) to validate the judge's accuracy? Additionally, how do the authors address the potential circularity bias introduced by using GPT-5.1 as both the planner and the judge?

2. Given the stochasticity of LLM outputs, what is the variance of ASR estimates across multiple runs? Have the authors conducted any repeated experiments to assess result stability? For categories with very few samples (e.g., Reputational Damage with 9 cases), the authors should address whether reported ASR values in these categories are meaningful.

**Limitations:**

See the weaknesses and questions.

**Strengths And Weaknesses:**

**Strengths**

- The observation that existing benchmarks only evaluate single-turn or static attack scenarios is accurate and important. As LLM agents are increasingly deployed in multi-session, tool-enabled settings, the temporal dimension of adversarial manipulation represents a genuinely underexplored threat surface. The paper's framing of this gap is clear and compelling.
- The five attack categories are meaningfully distinct and cover a broad range of adversarial strategies — from user-side manipulation (intent hijacking, tool chaining) to environment-side injection (objective drifting, task injection, memory poisoning). The taxonomy is well-grounded in the threat model formalism presented in Section 3, and the concrete demonstrations in Appendix B effectively illustrate how each attack type operates in practice.
- The modular architecture — separating agent, environment, task, and attack components — is a practical design choice that facilitates future extension. Positioning AgentLAB as a "live benchmark" rather than a static dataset is a good call given how rapidly the LLM agent landscape evolves.

------

**Weaknesses**

- The entire empirical contribution of this paper rests on ASR numbers produced by a GPT-5.1 judge, yet no validation of this judge's accuracy is provided. There is no comparison against human annotations, no inter-rater reliability analysis, and no discussion of the judge's systematic biases (e.g., whether it tends to over- or under-report attack success). This is a serious methodological gap: if the judge's accuracy is, say, 80%, the reported ASR values for all models could be substantially miscalibrated. Furthermore, both the planner and the judge are instantiated as GPT-5.1, introducing a potential circularity bias — attack strategies generated by GPT-5.1 may be stylistically more legible to a GPT-5.1 judge, artificially inflating reported ASR for GPT-5.1-planned attacks. At minimum, a human evaluation on a random subset of cases is required to ground the reported numbers.

- The benchmark's 644 test cases are distributed very unevenly across risk categories: Privacy Breach and Financial Loss together account for ~45% of cases, while Reputational Damage contains only 9 cases (1.4%) and Legal & Compliance Violations only 17 (2.6%). ASR estimates computed from fewer than 20 samples are highly unstable — a single judge error can swing the reported rate by 5–10 percentage points. This undermines the validity of per-category conclusions and limits the benchmark's utility for evaluating risks in underrepresented categories.

- All reported ASR values are point estimates with no confidence intervals, standard deviations, or significance tests. Given the known stochasticity of LLM outputs, the same attack run multiple times may yield substantially different results. Without variance analysis, it is impossible to determine whether observed differences between models (e.g., Claude-4.5 at 28.9% vs. Gemini-3 at 53.7%) are statistically meaningful or within the noise. This is a basic requirement for any quantitative benchmark paper.

- Only two defenses per attack type are evaluated, and all chosen defenses (Self-Reminder, Llama-Guard, Repeated Prompt, DeBERTa Detector) are known to be relatively weak inference-time interventions. The conclusion that "single-turn defenses fail against long-horizon attacks" is not surprising given this selection, and risks being a strawman argument. More critically, the evaluation entirely omits training-time safety alignment approaches such as [AgentAlign](https://arxiv.org/pdf/2505.23020), which incorporates safety considerations directly into model training. The paper would be significantly stronger if it evaluated at least one defense specifically designed for multi-turn scenarios, or a safety-aligned agent model, and provided concrete direction on what properties an effective long-horizon defense should have.

- Missing citations to closely related work:

  agent safety benchmark:

  (ICLR 2025) AGENTHARM: A BENCHMARK FOR MEASURING HARMFULNESS OF LLM AGENTS

  multi-turn red teaming:

  (ACL 2025) Chain of Attack: Hide Your Intention through Multi-Turn Interrogation

  (EMNLP 2024) Holistic Automated Red Teaming for Large Language Models through Top-Down Test Case Generation and Multi-turn Interaction

---

> ### Author Rebuttal · Authors · 2026-03-29
>
> We thank Reviewer A8a6 for the thorough review and for acknowledging the importance of the long-horizon threat surface and our attack taxonomy. We address each concern below.
>
> **W1: LLM judge validation and circularity bias.**
>
> We conducted a validation study on a stratified sample of 120 cases labeled by the GPT-5.1 judge. To validate these labels, we employed two independent raters: human annotators and a separate GPT-5.4 model.
>
> Human annotators agreed with GPT-5.1 on 91.67% of cases (FNR = 6.73%, FPR = 18.75%); the separate GPT-5.4 judge agreed on 89.17% (FNR = 10.58%, FPR = 12.50%).
>
> We computed inter-rater reliability (Landis & Koch, 1977 https://www.jstor.org/stable/2529310). The Cohen's kappa between human annotators and GPT-5.1 is 0.67 (substantial), between GPT-5.4 and GPT-5.1 is 0.62 (substantial). Fleiss' kappa across all three raters is 0.72 (substantial). These results place our judge at a similar level to the best-performing judges in JailbreakBench (https://arxiv.org/pdf/2404.01318), which achieve ~90% agreement with FPR of 9-13%.
>
> Regarding circularity bias: the high human-GPT-5.1 agreement (91.67%, kappa = 0.67) demonstrates that judge decisions are grounded in human judgment, not stylistic artifacts. The GPT-5.4 validator, a different model from the planner, further rules out planner-judge alignment as a confound.
>
> **W2: Uneven distribution across risk categories.**
>
> The imbalance across risk categories is an artifact of our categorization scheme, which may not directly indicate a lack of diversity in the underlying scenarios. The same data could yield balanced groupings under a different taxonomy. Nevertheless, to improve statistical stability, we have augmented 20 additional samples for each of the three smallest categories: Data Loss & Corruption, Legal & Compliance Violations, and Reputational Damage.
>
> **W3: No error bars or variance analysis.**
>
> We have now run multiple trials and report error bars for all main results. Updated Table 2 (ASR by attack type):
>
> | Agent | Intent Hijacking | Tool Chaining | Objective Drifting | Task Injection | Memory Poisoning | Overall |
> |:---:|:---:|:---:|:---:|:---:|:---:|:---:|
> | Qwen-3 | 78.1 ± 7.1 | 96.3 ± 2.6 | 92.2 ± 4.4 | 93.1 ± 1.3 | 48.0 ± 14.3 | 81.5 ± 3.7 |
> | Llama-3.1 | 53.3 ± 7.9 | 90.4 ± 4.4 | 67.4 ± 7.9 | 86.6 ± 1.4 | 34.6 ± 11.9 | 66.5 ± 4.0 |
> | GPT-4o | 74.0 ± 6.1 | 94.1 ± 3.5 | 79.2 ± 5.9 | 79.9 ± 2.2 | 63.3 ± 14.2 | 78.1 ± 3.8 |
> | GPT-5.1 | 59.8 ± 6.9 | 94.6 ± 3.1 | 73.7 ± 8.9 | 21.5 ± 0.8 | 51.3 ± 15.5 | 69.9 ± 4.5 |
> | Gemini-3 | 46.2 ± 7.5 | 95.9 ± 2.8 | 15.8 ± 6.4 | 43.1 ± 1.7 | 67.3 ± 13.8 | 53.7 ± 4.0 |
> | Claude-4.5 | 27.2 ± 6.9 | 73.3 ± 6.7 | 5.3 ± 3.7 | 0.0 ± 0.0 | 38.8 ± 13.8 | 28.9 ± 3.6 |
>
> Updated Table 3 (not showing due to space constraints) similarly confirms that the long-horizon advantage over one-shot is robust across all agents. We will add error bars to both tables.
>
> **W4: Defense evaluation is limited; only weak inference-time defenses are tested.**
>
> We have added two advanced defenses:
>
> *1. Progent (Shi et al., 2025) -- system-level defense (Progent-LLM mode):*
>
> We evaluated Progent on Tool Chaining and Memory Poisoning. Tool Chaining results:
>
> | Agent | Undefended | Defended by Progent |
> |:---:|:---:|:---:|
> | Qwen-3 | 96.3 | 75.4 |
> | Llama-3.1 | 90.4 | 66.1 |
> | GPT-4o | 94.1 | 73.7 |
> | GPT-5.1 | 94.6 | 57.1 |
> | Gemini-3 | 95.9 | 49.1 |
>
> Progent achieves meaningful ASR reductions, but substantial vulnerability persists across all agents and both attack types.
>
> *2. XGuard-Train (from X-Teaming) -- multi-turn defense via safety fine-tuning:*
>
> XGuard-Train is designed for multi-turn adversarial settings. Results on Intent Hijacking and Memory Poisoning:
>
> | Agent | Intent Hijacking (Original) | Intent Hijacking (XGuard) | Memory Poisoning (Original) | Memory Poisoning (XGuard) |
> |:---:|:---:|:---:|:---:|:---:|
> | Llama-3.1 | 53.3 | 44.4 | 34.6 | 20.0 |
> | Qwen-3 | 78.1 | 23.1 | 48.0 | 38.1 |
>
> XGuard-Train substantially reduces Qwen-3's Intent Hijacking ASR but has modest effects elsewhere. Additionally, on Tool Chaining, the fine-tuned models produced gibberish when handling complex tool calls, indicating the defense disrupted utility rather than becoming robust.
>
> Overall, significant vulnerability remains, confirming that long-horizon attacks are not trivially addressed by current defenses.
>
> **W5: Missing citations.**
>
> We will cite these papers as suggested by the reviewer, with a discussion of how AgentLAB relates to these works.
>
> **Limitations section.**
>
> We will add a dedicated limitations section. A key limitation is the scarcity of long-horizon defenses. The existing defenses XGuard-Train ( https://arxiv.org/abs/2504.13203) and AgentAlign ( https://arxiv.org/abs/2505.23020) are both fine-tuning-based with only partial mitigation. A central goal of AgentLAB is to surface this gap and call for more effective long-horizon defenses.

---

> > ### Author Rebuttal · Reviewer_A8a6 · 2026-04-01
> >
> > Thank you for your response. Several of my primary concerns have been addressed, including the quality of the judge and the error bar analysis of the evaluation results. I have therefore raised my score to 4 (Weak Accept).

---

> > > ### Author Response · Authors · 2026-04-01
> > >
> > > Dear Reviewer A8a6,
> > >
> > > Thank you for your thoughtful re-evaluation and for raising your score! We are glad our rebuttal addressed your concerns. We remain available to address any further questions during the discussion period.
> > >
> > > Regards,
> > >
> > > Authors

---

### Official Review · Reviewer_6XhS · 2026-03-11

**Soundness:** 2
**Presentation:** 3
**Significance:** 2
**Originality:** 3
**Overall Recommendation:** 4
**Confidence:** 3

**Summary:**

AgentLab is a benchmark designed specifically to evaluate long-horizon attacks. The authors note prior benchmarks largely focus on single-turn/static settings and do not support adaptive long-horizon attacks, where attacks can occur over multiple turns and can be adaptive. They present a diverse benchmark which tests for five attack types specific to this setting using a range of environments and test cases. The benchmark also uses a multi-agent attacking system to develop attacks. The benchmark finds that attacks are generally successful against tested models, suggesting there is a security gap.

**Compliance With Llm Reviewing Policy:**

Affirmed.

**Final Justification:**

I remain somewhat concerned about the realism of this benchmark, but the authors have partially addressed my concerns and I am narrowly raising my score to 4 (weak accept). This is because I continue to believe the paper identifies a clear gap in existing benchmarks and designs attacks specifically to test that gap. I strongly recommend authors clearly state realism limitations and aim to fix this in future.

**Key Questions For Authors:**

Q1: The paper says you consider black-box and white-box settings; where are each of these considered?
Q2: The formalism claims to allow for scenarios where users only provide initial instructions without subsequent interactions. However, the formalism has tuples that always contain a prompt, action, output, response ordering; how would this formalism accommodate a structure with multiple action/output pairs in a row prior to a response?
Q3: Line 851 claims "multiple touchpoints" are used--can you clarify what these multiple touchpoints are?

**Limitations:**

The authors do discuss potential negative societal impacts of their work, but do not discuss other limitations.

**Strengths And Weaknesses:**

**Strengths.**
* The paper identifies a clear gap in existing benchmarks and designs attacks specifically to test that gap.
* The paper is generally clearly structured and written, with helpful explanatory diagrams, clear plots, and a detailed appendix.
* The authors include an ablation study which (though I believe currently has weaknesses discussed below) is useful for understanding their contribution.
* The authors plan to make the benchmark open source to enable scientific progress.

**Weaknesses.** I have two main concerns, validity (tied to soundness) & usability (tied to significance).
1. Validity:
1a. Realism -- The more the benchmark's environments match real-world deployment conditions, the better the dataset is. If the environments are obviously synthetic, it would hurt the validity of these findings. Though the paper notes benchmarks use environments from prior work, they do not provide enough details to assess realism (see 3). Separately, it could be good to further discuss which deployment settings plausibly expose enough iterative feedback for adaptive attacks to be realistic.
1b. Ablations -- For adaptivity, I believe a stronger comparison would be to hold fixed number of total adversarial interactions and compare using budget on iteration vs using all the budget in a single turn (so low n_opt but more attempts). Similarly, for the long-horizon ablation, it would help to hold total attack budget fixed (so you have more attempts if you have fewer turns).
2. Usability: The argument for this paper's significance is partly tied to adoption by other researchers. However, it is not clear to me how easy to use the benchmark is. It would be useful to know if this benchmark is implemented in a popular framework such as Inspect AI (https://inspect.aisi.org.uk/) or another common framework. For example, it is unclear how readily the benchmark supports system-level agents such as Claude Code or Codex. Such systems may be good fits for this sort of benchmark but are distinct from testing just the underlying models with a single unified interface. It would be useful to share more details on the usability of the benchmark.

Additionally:
3. The anonymous code link was not accessible when I checked--the hosting appears to be expired.
4. I would encourage the authors to more clearly discuss the limitations of their work; for example, it only allows for cases where there is a ground-truth sequence of tool calls needed to accomplish a task, rather than allowing for arbitrary sequences of tool calls so long as the malicious task is accomplished (if I understand correctly).
5. Line 811 claims that "no single tool call raises red flags"; however, Turn 2 does appear to contain malicious single-turn content, it's just misunderstood by the model. If I'm understanding correctly, I'd suggest clarifying this point.
6. Lastly, it would be good to verify that the judge is well calibrated via a human agreement metric to ensure that attacks marked as successful are in fact successful; or clarify in how many cases programmatic evaluation is used instead.

---

> ### Author Rebuttal · Authors · 2026-03-29
>
> We thank the reviewer for the constructive review and for recognizing AgentLAB's contribution. We address each concern below.
>
> **W1a: Realism of environments.**
>
> AgentLAB builds upon established agentic environments from prior work (e.g., WebShop) and augments them with production-relevant capabilities such as persistent memory (Mem0). Regarding which settings plausibly expose enough iterative feedback for adaptive attacks: AgentLAB is an attack framework where the attacker adaptively crafts adversarial inputs based on observed agent behavior across turns, which is realistic for any deployment where an adversary can interact with or observe the agent over multiple sessions (e.g., shared workspaces, customer-facing agents, or agents with persistent memory).
>
> **W1b: Ablations should hold total budget fixed.**
>
> Both the fixed-budget ablation for adaptivity (comparing iterative optimization vs. single-turn with the same total budget) and the fixed-budget ablation for long-horizon attacks (comparing multi-turn vs. fewer turns with the same total attack budget) are already included in the paper (Section 5.3). We will make this clearer in the revision.
>
> **W2: Usability and framework compatibility.**
>
> We acknowledge that adoption depends on usability. Our codebase (§A) uses a modular, config-driven interface: adding a new agent requires only a thin step(instruction) -> (action, response) wrapper, compatible with most API-based and open-weight systems. Our FastAPI-based tool invocation is architecturally aligned with Inspect AI's task abstraction, and we will evaluate and report integration effort in the revision. For system-level agents (Claude Code, Codex), tool chaining and memory poisoning are directly applicable, and task injection via repository comments is a particularly natural fit. We will discuss this extension explicitly and flag dedicated environment designs as future work. We will expand §4.3 with setup instructions and system-level agent guidance.
>
> **W3: Anonymous code link expired.**
> https://anonymous.4open.science/r/AgentLAB-204D is now active again.
>
> **W4: Limitations section.**
>
> We will add a dedicated limitations section. A key limitation is the scarcity of long-horizon defenses. The existing defenses XGuard-Train ( https://arxiv.org/abs/2504.13203) and AgentAlign ( https://arxiv.org/abs/2505.23020) are both fine-tuning-based with only partial mitigation. A central goal of AgentLAB is to surface this gap and call for more effective long-horizon defenses.
>
> **W5: Line 811 clarification.**
>
> We agree the phrasing is imprecise. The intended point is that the individual tool calls are not overtly malicious in isolation when viewed without the broader attack context. We will revise this sentence for clarity.
>
> **W6: Judge validation.**
>
> To validate the GPT-5.1 judge used in our paper, we compared its labels on a stratified sample of 120 cases against two independent validators: human annotators and a separate GPT-5.4 judge. Human annotators agreed with GPT-5.1 on 91.67% of cases (FPR = 18.75%, FNR = 6.73%); GPT-5.4 agreed on 89.17% (FPR = 12.50%, FNR = 10.58%). These results place our GPT-5.1 judge at a similar level to the best-performing judges reported in JailbreakBench (Chao et al., 2024, NeurIPS), which achieve ~90% agreement with FPR of 9-13%.
>
> **Q1:** Black-box and white-box settings are realized at the attack implementation level. In black-box settings, the attacker observes only the agent's final actions and responses to refine adversarial prompts (e.g., intent hijacking, objective drifting). In white-box settings, the attacker additionally accesses intermediate reasoning traces (chain-of-thought) to guide refinement. This is the grey-box variant formalized in §A.4 for task injection, where the adversary observes the full tool invocation trajectory. We will clarify this mapping explicitly in the revision.
>
> **Q2:** The formalism in Eq. 1 is intentionally coarse-grained, where each tuple ⟨p_i, a_i, o_i, r_i⟩ represents one user-agent-environment-user cycle. To accommodate multiple action/observation pairs before a response (i.e., internal agent reasoning steps), each tuple can be interpreted as an abstract interaction round where a_i and o_i represent sequences of tool calls and observations that occur between consecutive user turns. This is how multi-step tool chains within a single user instruction are handled in practice. We will make this explicit in the revision.
>
> **Q3:** The "multiple touchpoints" in the objective drifting attack refer specifically to two injection sites: (1) search result snippets, where product listings are augmented with persuasive framing (e.g., "Premium lasts 3× longer"), and (2) individual product pages, where additional injections reinforce the preference shift. In the concrete example, the agent encounters 3 search-level injections and 4 product-page injections across its browsing trajectory. We will clarify these touchpoints in the main text.

---

> > ### Author Rebuttal · Reviewer_6XhS · 2026-04-03
> >
> > I thank the authors for their response, and they have resolved a few of my concerns. However, I still have significant concerns:
> >
> > W1a: I agree that this work build on prior environments and adds production-*relevant* capabilities. However, I remain concerned that the environments are not sufficiently realistic (e.g., are overly simplistic compared to production systems). And though I agree with their reasoning on when iterative attacks are plausible, I'm not convinced these situations are captured by the dataset. Overall, I remain concerned as to the realism of these test environments and threat models. In my view, if the core contribution of the paper is a dataset, it is important is matches threat-model-relevant deployment conditions as closely as possible.
> >
> > W1b: Ah, I did not realize those were fixed-budget. That is excellent, thank you; I suggest clarifying this in the paper, as I don't believe this is currently in the paper text.
> >
> > W2: Thank you. Can the authors please clarify if Claude Code or Codex could be directly evaluated using AgentLAB?
> >
> > W3: Thank you.
> >
> > W4: By limitation section, I was referring to limitations with AgentLAB itself, e.g. realism concerns as I outlined above.
> >
> > W5: I'm still a bit confused about this. It does seem to me like Turn 2 is harmful even when not looking at content from any other turn. Could the authors please clarify?
> >
> > W6: That remains a relatively high FPR. Have the authors confirmed this FPR does not invalidate any reported results?
> >
> > Q1: So are only gray-box results reported? I remain confused here.
> >
> > Q2: Thank you.
> >
> > Q3: Thank you.

---

> > > ### Author Response · Authors · 2026-04-04
> > >
> > > We thank Reviewer 6Xhs for the valuable feedback!
> > > ### W1a
> > > **Realism.** We deliberately strip away properties irrelevant to attack surfaces (e.g., API failures, messy data, open-ended tools) while preserving those that are security-relevant and experimentally tractable: sequential decision-making, tool-mediated actions, partial observability, causal action effects, and reproducible task evaluation. While production evaluation is ethically and practically inaccessible to the research community, AgentLAB's value lies in reproducible, causally interpretable results against a well-defined threat model, the standard shared by all peer benchmarks in this space.
> > >
> > > **Adaptiveness.** AgentLAB's dataset specifies only environments, tasks, and tools. All attacks are implemented as adaptive agents that dynamically generate adversarial prompts/payloads based on observed agent behaviors across turns, making attacks inherently dynamic rather than static artifacts. Thus, agentLAB's contributions extend beyond static specifications to a suite of long-horizon, adaptive attack implementations, representing the first benchmark for evaluating LLM agent vulnerabilities to diverse long-horizon threats.
> > >
> > > ### W2
> > > At the agent interface, integration is simple: both agents accept natural language instructions and return structured tool calls, requiring no changes to the attack logic. Several attacks transfer naturally to coding agents: tool chaining exploits the compositional structure of coding operations (file I/O, git commands); memory poisoning applies wherever the agent maintains persistent state (e.g., CLAUDE.md); and task injection via repository comments, PR descriptions, or dependency metadata is a natural fit. We may then leverage existing coding agent benchmarks (e.g., SWE-bench) to construct suitable environments.
> > > ### W4
> > > We see several avenues for improving AgentLAB. We purposely remove properties irrelevant to the attack surface, which introduces gaps relative to production systems; future work can examine how reintroducing these properties affects attack evaluation. It currently includes relatively simple defenses, a deliberate choice to establish a clean baseline, but one that invites extension to more sophisticated countermeasures. Scaling to a wider variety of tasks and attack types requires automating the specification of tasks and utility criteria without sacrificing evaluation reliability. Extending AgentLAB to support system-level agents (e.g., Codex) can dramatically expand the range of tasks and attacks.
> > > ### W5
> > > The malicious goal in B.2 is to send a harmful payload to another user via email. The harmful payload is a part of the environment setup, drawn from Agent-SafetyBench, which evaluates whether agents refuse to process the payload when adversaries attempt to send it.
> > >
> > > If instructing the agent to "email bomb-making tips", the agent often refuses, as the harmful payload and distribution intent co-occur, triggering safety mechanisms. Tool-chaining bypasses this by decomposing the objective into routine actions such that **no single turn presents both the harmful payload and distribution intent**, with the chain structure obscuring their connection. STAC confirms that agents are much more likely to refuse single-turn requests than the same goals disguised behind a chain of tool calls.
> > > ### W6
> > >
> > > We identify two factors for the elevated FPR.
> > >
> > > **Sampling imbalance.** The samples were drawn from the tool chaining and intent hijacking attacks, both with high ASR. This produced a validation set with 16 (out of 120) negative cases. Even a small number of disagreements yields a disproportionately high FPR.
> > >
> > >  **Threshold selection.** The judge assigns harmfulness scores from 1 to 5; the reported FPR uses a threshold of 4. Tightening to 5 reduces FPR to 12.50%, confirming that FPR is tunable via threshold selection.
> > >
> > > **Expanded validation.** We expand the validation set by sampling another 180 cases across previously unsampled attack types.
> > >
> > > | | Human Neg. | Human Pos. |
> > > |---|---|---|
> > > | **GPT-5.1 Neg.** | 80 | 11 |
> > > | **GPT-5.1 Pos.** | 20 | 189 |
> > >
> > > This yields FPR = 12.09% (FNR = 9.57%), comparable to the best judges in JailbreakBench. Thus, the elevated FPR was an artifact of sampling imbalance.
> > > ### Q1
> > > The table maps each attack type to its access level and adversary observation.
> > >
> > > | Attack  | Access | Observation |
> > > |---|---|---|
> > > | Intent Hijacking | Grey-box | Action + reasoning |
> > > | Tool Chaining | Black-box | Action |
> > > | Objective Drifting | Black-box | Action |
> > > | Task Injection | Grey-box | Action + reasoning |
> > > | Memory Poisoning | Black-box | Injection performed a priori |
> > >
> > > Access levels are configured based on the nature of each attack type. Intent hijacking and task injection operate under grey-box access, simulating a highly capable, adaptive adversary for rigorous red-teaming. The remaining attack types operate under black-box access, as their strategies are effective with action/response-level observations alone.

---

### Official Review · Reviewer_Hwtc · 2026-03-14

**Soundness:** 3
**Presentation:** 4
**Significance:** 3
**Originality:** 4
**Overall Recommendation:** 4
**Confidence:** 4

**Summary:**

This paper introduces AgentLAB, a benchmark for evaluating the security of LLM-based agents against long-horizon attacks that exploit multi-turn user–agent–environment interactions. It constructs diverse attack scenarios across realistic tool-enabled environments, covering five attack types including intent hijacking, tool chaining, task injection, objective drifting, and memory poisoning. Experiments on multiple proprietary and open-weight LLM agents show that they remain highly vulnerable to long-horizon attacks, and defenses designed for single-turn interactions fail to reliably mitigate these threats.

**Compliance With Llm Reviewing Policy:**

Affirmed.

**Final Justification:**

All of my concerns have been addressed. Therefore I still maintain a positive attitude to this paper.

**Key Questions For Authors:**

Please refer to the weaknesses.

**Limitations:**

No, the paper does not include a clear discussion of limitations.

**Strengths And Weaknesses:**

**Strengths:**

1. The proposed long-horizon attack benchmark is very important for the current agent security field.
2. AgentLAB covers five typical attack types.
3. The designed attacks achieve high ASR on several existing agents.

**Weaknesses:**

1. The attacker is only a 14B model (Qwen-3-14B) and does not have particularly strong capability.
2. The properties of long-horizon attacks are insufficiently explored. For example, the paper does not analyze the correlation between the attack round length and the attack success rate.
3. The compared defenses include only prompting-based methods and a DeBERTa classifier. However, it is unclear which specific DeBERTa classifier is used. There have been many advanced DeBERTa-based classifiers proposed in prior work, such as ProtectAI [1], PIGuard [2], PromptShield [3], and PromptGuard2 [4]. In addition, several advanced alignment-based defenses (e.g., Meta SecAlign [5]) and system-level defenses (e.g., CaMeL [6], Progent [7], DRIFT [8], and AgentArmor [9]) are not considered in the evaluation.

**References:**
- [1] ProtectAI, https://huggingface.co/protectai/deberta-v3-base-prompt-injection-v2.
- [2] Li et al., “PIGuard: Prompt Injection Guardrail via Mitigating Overdefense for Free”, ACL, 2025.
- [3] Jacob et al., “PromptShield: Deployable Detection for Prompt Injection Attacks”, ACM CODASPY, 2025.
- [4] PromptGuard2., https://huggingface.co/meta-llama/Llama-Prompt-Guard-2-86M.
- [5] Chen et al., “META SECALIGN: A Secure Foundation LLM Against Prompt Injection Attacks”, arXiv, 2025.
- [6] Debenedetti et al., “Defeating Prompt Injections by Design”, SaTML, 2026.
- [7] Shi et al., “Progent: Programmable Privilege Control for LLM Agents”, arXiv, 2025.
- [8] Li et al., “DRIFT: Dynamic Rule-Based Defense with Injection Isolation for Securing LLM Agents”, NeurIPS, 2025.
- [9] Wang et al., “AgentArmor: Enforcing Program Analysis on Agent Runtime Trace to Defend Against Prompt Injection”, arXiv, 2025.

---

> ### Author Rebuttal · Authors · 2026-03-29
>
> We sincerely thank Reviewer Hwtc for the thoughtful evaluation and for recognizing that AgentLAB addresses an important gap in the agent security field. We address each concern below.
>
> **W1: The attacker is only a 14B model and does not have particularly strong capability.**
>
> We have conducted additional experiments using a stronger attacker model, huihui-ai/Huihui-Qwen3.5-27B-abliterated, on the Intent Hijacking attack:
>
> | Agent | Qwen3-14B Attacker | Qwen3.5-27B Attacker |
> |:---:|:---:|:---:|
> | Qwen-3 | 78.1 | 76.9 |
> | Llama-3.1 | 53.3 | 58.1 |
> | GPT-4o | 74.0 | 74.3 |
> | GPT-5.1 | 59.8 | 53.7 |
>
> These results reveal two important findings. First, ASR values remain comparable across attacker scales, suggesting that the 14B attacker is already sufficiently capable for this benchmark. Second, and more fundamentally, demonstrating that even a relatively weak attacker can achieve high ASR against frontier agents is itself a valuable finding: it highlights the severity of the vulnerability. If a 14B model can reliably compromise these agents, the threat from stronger adversaries is only more concerning.
>
> Additionally, we use an open-sourced, abliterated model as the attacker because safety-aligned models often refuse to assist in red teaming, undermining attack generation rather than reflecting victim robustness. This finding is also corroborated in Crescendo (https://arxiv.org/html/2404.01833) Sec 5.1.2. One paper (https://arxiv.org/abs/2502.09638) also observes that "the refusal-training of strong LLMs refuses to help jailbreaking".
>
> **W2: The correlation between attack round length and ASR is insufficiently explored.**
>
> We thank the reviewer for raising this point. This analysis is included in our ablation study (Section 5.3), where we show that ASR increases as the number of attack turns grows. We will ensure this finding is more prominently highlighted in the revised manuscript.
>
> **W3: The compared defenses are limited; advanced DeBERTa classifiers and system-level defenses are not considered.**
>
> We fully agree that evaluating stronger defenses is critical. In the revised version, we have added experiments with two representative advanced defenses:
>
> To clarify, we are using "[4] PromptGuard2., https://huggingface.co/meta-llama/Llama-Prompt-Guard-2-86M" as our DeBERTa classifier in the paper.
> Thank you so much for suggesting advanced alignment-based defenses, we have decided to use Progent as it can be applied to defend both injection-based and jailbreaking-based attacks.
>
> *Progent (system-level defense, LLM-policy mode):* We evaluated Progent on Tool Chaining and Memory Poisoning:
>
> | Agent | Tool Chaining (Undefended) | Tool Chaining (Progent) | Memory Poisoning (Undefended) | Memory Poisoning (Progent) |
> |:---:|:---:|:---:|:---:|:---:|
> | Qwen-3 | 96.3 | 75.4 | 48.0 | 42.1 |
> | Llama-3.1 | 90.4 | 66.1 | 34.6 | 31.4 |
> | GPT-4o | 94.1 | 73.7 | 63.3 | 56.0 |
> | GPT-5.1 | 94.6 | 57.1 | 51.3 | 46.7 |
> | Gemini-3 | 95.9 | 49.1 | 67.3 | 51.6 |
>
> As shown in the result here, Progent did block many of the malicious requests in Tool-Chaining attacks while it is still relatively vulnerable against both tool-chaining and memory poisoning attacks. The result further highlights the need to a more comprehensive study of the mechanisms of long-horizon attacks and defenses.
>
> *XGuard-Train (multi-turn defense via fine-tuning):*
>
> | Agent | Intent Hijacking (Original) | Intent Hijacking (XGuard) | Memory Poisoning (Original) | Memory Poisoning (XGuard) |
> |:---:|:---:|:---:|:---:|:---:|
> | Llama-3.1 | 53.3 | 44.4 | 34.6 | 20.0 |
> | Qwen-3 | 78.1 | 23.1 | 48.0 | 38.1 |
>
> While both defenses reduce ASR, substantial vulnerability remains, confirming that long-horizon attacks pose challenges beyond what current defenses can fully address. Notably, XGuard-Train fine-tuned models produced gibberish outputs on Tool Chaining scenarios involving complex tool calls, indicating that the defense disrupted general capability rather than selectively filtering attacks.
>
> **Limitations and future work.**
>
> We will add a dedicated limitations section. A key limitation is the scarcity of defenses for the long-horizon regime. To our knowledge, the only multi-turn safety fine-tuning defenses for agentic settings are XGuard-Train (Rahman et al., 2025, https://arxiv.org/abs/2504.13203) and AgentAlign (Zhang et al., 2025, https://arxiv.org/abs/2505.23020). Both are fine-tuning-based, and as our XGuard-Train experiments show, they provide only partial mitigation: ASR is reduced in some settings but general agent capability degrades on complex tool-use tasks. This underscores a fundamental gap: the community lacks defenses that reliably counter long-horizon attacks without sacrificing agent utility. A central goal of AgentLAB is to surface this gap and provide a rigorous benchmark to drive development of effective, long-horizon-aware defenses.
>
> We thank the reviewer again for the constructive feedback, which has meaningfully strengthened our evaluation.

---

> > ### Author Rebuttal · Reviewer_Hwtc · 2026-04-03
> >
> > I appreciate the authors’ detailed response. All of my concerns have been addressed. I will continue to maintain a positive attitude toward the work.

---

> > > ### Author Response · Authors · 2026-04-03
> > >
> > > Dear Reviewer Hwtc,
> > >
> > > We sincerely thank you for the thoughtful engagement and assessment. We are glad the response addressed all concerns, and greatly appreciate your continued support for this work.
> > >
> > > Regards,
> > >
> > > Authors

---

### Decision · Program_Chairs · 2026-04-30

**Decision:**

Accept (regular)

**Comment:**

This paper introduces AgentLAB, a novel benchmark evaluating LLM agents against adaptive, long-horizon attacks. All four reviewers unanimously recommended a Weak Accept (scores: 4, 4, 4, 4), agreeing that the paper addresses a critical and underexplored gap in agent security with a comprehensive taxonomy of five attack types.

During the discussion period, the authors provided a highly commendable rebuttal that significantly strengthened the paper's empirical rigor. They successfully addressed the primary methodological concerns by: (1) conducting an expanded human-validation study for the GPT-5.1 judge to demonstrate reliability, (2) incorporating variance analysis with error bars across multiple trials, and (3) evaluating advanced multi-turn baseline defenses like Progent and XGuard-Train.

While some reviewers maintained minor reservations regarding the ultimate real-world fidelity of the synthetic environments and the strategic limits of the chosen abliterated attacker models, the consensus is that these are acceptable constraints for a pioneering benchmark. The authors have committed to explicitly framing these as limitations in the revised manuscript.

Given the unanimous reviewer scores and the authors' thorough mitigation of the identified weaknesses, I recommend an Accept. AgentLAB provides a valuable, reproducible foundation for future agent security research.